# Long Non-Coding RNAs Associated with Mitogen-Activated Protein Kinase in Human Pancreatic Cancer

**DOI:** 10.3390/cancers15010303

**Published:** 2023-01-02

**Authors:** Tomohiko Ishikawa, Shinichi Fukushige, Yuriko Saiki, Katsuya Hirose, Takako Hiyoshi, Takenori Ogawa, Yukio Katori, Toru Furukawa

**Affiliations:** 1Department of Investigative Pathology, Tohoku University Graduate School of Medicine, Sendai 980-8575, Japan; 2Department of Otolaryngology-Head and Neck Surgery, Tohoku University Graduate School of Medicine, Sendai 980-8574, Japan; 3Department of Otolaryngology, Gifu University Graduate School of Medicine, Gifu 501-1194, Japan

**Keywords:** lincRNA, pancreatic cancer, MAPK, signal transduction, promoter, transcriptome

## Abstract

**Simple Summary:**

Long non-coding RNAs (lncRNAs) have emerged as a significant player in various cancers, including pancreatic cancer. However, how lncRNAs are aberrantly expressed in cancers is largely unknown. We hypothesized that lncRNAs would be regulated by signaling pathways and contribute to malignant phenotypes of cancer. In this study, to understand the significance of mitogen-activated protein kinase/extracellular signal-regulated kinase (MAPK/ERK) for the expression of lncRNAs in pancreatic cancer, we performed comparative transcriptome analyses between pancreatic cancer cell lines with or without activation of MAPK. We identified 45 lncRNAs presumably associated with MAPK in pancreatic cancer cells; among these, LINC00941 was consistently upregulated by MAPK. The promoter of LINC00941 was determined and found to be preferentially associated with MAPK activity via ETS-1 binding site. TCGA data analysis indicated that high expression of LINC00941 was associated with poor prognosis of pancreatic cancer patients. Downstream targets of LINC00941 were involved in 44 biological processes, including the glycoprotein biosynthetic process, beta-catenin-TCF complex assembly, and histone modification. These results indicate that MAPK mediates the aberrant expression of lncRNAs, and LINC00941 is the most consistently promoted lncRNA by MAPK in pancreatic cancer. Therefore, MAPK-associated lncRNAs might represent both potentially valid therapeutic targets and diagnostic biomarkers.

**Abstract:**

Long non-coding RNAs (lncRNAs) have emerged as a significant player in various cancers, including pancreatic cancer. However, how lncRNAs are aberrantly expressed in cancers is largely unknown. We hypothesized that lncRNAs would be regulated by signaling pathways and contribute to malignant phenotypes of cancer. In this study, to understand the significance of mitogen-activated protein kinase/extracellular signal-regulated kinase (MAPK/ERK), which is a major aberrant signaling pathway in pancreatic cancer, for the expression of lncRNAs, we performed comparative transcriptome analyses between pancreatic cancer cell lines with or without activation of MAPK. We identified 45 lncRNAs presumably associated with MAPK in pancreatic cancer cells; among these, LINC00941 was consistently upregulated by MAPK. The immediate genomic upstream region flanking LINC00941 was identified as a promoter region, the activity of which was found to be preferentially associated with MAPK activity via ETS-1 binding site. LINC00941 promoted cell proliferation in vitro. Moreover, TCGA data analysis indicated that high expression of LINC00941 was associated with poor prognosis of patients with pancreatic cancer. Transcriptomes comparing transcriptions between cells with and without LINC00941 knockdown revealed 3229 differentially expressed genes involved in 44 biological processes, including the glycoprotein biosynthetic process, beta-catenin-TCF complex assembly, and histone modification. These results indicate that MAPK mediates the aberrant expression of lncRNAs. LINC00941 is the lncRNA by MAPK most consistently promoted, and is implicated in the dismal prognosis of pancreatic cancer. MAPK-associated lncRNAs may play pivotal roles in malignant phenotypes of pancreatic cancer, and as such might represent both potentially valid therapeutic targets and diagnostic biomarkers.

## 1. Introduction

Pancreatic cancer is one of the leading causes of cancer death [1,2]. Despite advancements in diagnostic and therapeutic techniques, the 5-year survival rate of patients with pancreatic cancer remains less than 10% [1,2]. This dismal prognosis highlights an urgent need to develop sensitive diagnostic biomarkers as well as effective therapeutics. It is essential to understand the molecular pathogenesis of cancer in order to develop a molecular-oriented method of diagnosis and/or targeted therapy. Pancreatic cancer is characterized by constitutive activation of the mitogen-activated protein kinase/extracellular signal-regulated kinase (MAPK/ERK) pathway provoked by the prevailed *KRAS* mutation. The activated MAPK translocates into the nucleus, phosphorylate various transcription factors, and alter the expression of downstream genes, which leads to diverse cellular responses [3]. The MAPK pathway is recognized as an important target for targeted molecular drugs for which many drugs have been developed and clinically applied, such as sorafenib, trametinib, cobimetinib, and selumetinib, although unfortunately these drugs have not proven clinically effective for pancreatic cancer [4,5]. To overcome this refractoriness and to develop efficient therapies for pancreatic cancer, more knowledge of pathobiological effects of MAPK activation is needed.

Long non-coding RNAs (lncRNAs) are non-coding RNAs greater than 200 nucleotides [6]. Recent advancements in high-throughput genomic technologies have elucidated the implications of previously unannotated non-protein coding transcripts, namely, small nuclear RNAs, small nucleolar RNAs, microRNAs (miRNAs), and lncRNAs, in various disease settings. LncRNAs have been recognized as critical regulators for various biological processes, including DNA replication, transcription, RNA splicing, translation, and epigenetic regulation [6,7]. Moreover, many studies have demonstrated that lncRNAs play important roles in the development and progression of various cancers, including pancreatic cancer, and the aberrant expression of lncRNAs can be a prognostic biomarker [6,7]. However, the mechanism of aberrant expression of lncRNAs in cancer is largely unknown. The expression of genes is primarily controlled by the functions of signal transduction pathways. We previously uncovered that a number of miRNAs are downstream targets of MAPK, and that these are implicated in cell proliferation and survival by modulating certain effector genes in pancreatic cancer [8]. Furthermore, these miRNAs represent useful serum biomarkers to discriminate pancreatic cancer from benign diseases [9]. Hence, we hypothesized that the MAPK signaling pathway would play a significant role in the aberrant expression of lncRNAs involved in pancreatic cancer. In this study, we explore lncRNAs associated with MAPK activity in pancreatic cancer cells, their clinical impacts, and the mechanism of their implications.

## 2. Materials and Methods

### 2.1. Methodological Flowchart

A methodological flowchart of this study is depicted in Appendix A.

### 2.2. Cell Lines

Established human pancreatic cancer cell lines MIA PaCa-2, AsPC-1, and PCI-35 were used in this study. MIA PaCa-2 and AsPC-1 were obtained from American Type Culture Collection (Manassas, VA, USA) and cultured as instructed by the supplier. PCI-35 was provided by Dr. Hiroshi Ishikura and cultured as described in the original publication [10].

### 2.3. Inhibition of MAPK

The pancreatic cancer cells were seeded in 100-mm culture dishes at 5 × 10^5^ cells per dish with an appropriate culture medium containing 10% FBS, then cultured under 37 °C in 5% CO_2_ with the appropriate humidity. After 24 h, the medium was replaced with a medium containing U0126 (Sigma-Aldrich, St. Louis, MO, USA), a MAP2K inhibitor, dissolved in dimethyl sulfoxide (DMSO) at a final concentration of 10 μmol/L, or replaced with a medium containing the same volume of DMSO without U0126 as a control. After 24 h, RNA isolation and protein extraction were carried out.

### 2.4. Immunoblotting

Cells were lysed in RIPA buffer (Sigma-Aldrich) supplemented with inhibitors of proteinases and phosphatases (Complete Mini and PhosSTOP; Roche Diagnostics, Rotkreuz, Switzerland). The cell lysate with 20 μg of protein was electrophoresed in a 10–20% gradient polyacrylamide gel (DRC, Tokyo, Japan) and transferred to Clear Blot Membrane-p (ATTO, Tokyo, Japan). The blot was blocked using PBS with 0.1% Tween-20 containing 2% bovine serum albumin or 0.8% ECL Prime Blocking Reagent (GE Healthcare, Chicago, IL, USA). Primary antibodies employed were monoclonal anti-MAP kinase, activated (diphosphorylated ERK1 and 2) (clone MAPK-YT; 1:3000; Sigma-Aldrich), a monoclonal anti-p44/42 MAPK (Erk1/2) (clone 137F5; 1:1500; Cell Signaling Technology, Danvers, MA, USA), and monoclonal anti-beta actin (clone AC-15; 1:2000; Sigma-Aldrich). The secondary antibodies employed were a horseradish peroxidase (HRP)-conjugated anti-mouse immunoglobulin (1:30,000; GE Healthcare) or an HRP-conjugated anti-rabbit immunoglobulin (1:10,000; Cell Signaling Technology). These antibodies were diluted with Can Get Signal (TOYOBO, Osaka, Japan) according to the manufacturer’s instructions. Signals were visualized by the reaction with ECL Prime Detection Reagent (GE Healthcare) and digitally processed using an LAS 4000 mini-CCD camera system (Fuji Photo Film, Tokyo, Japan).

### 2.5. Whole Transcriptome Sequencing

Total RNA was isolated using the RNeasy Mini Kit (QIAGEN, Hilden, Germany) according to the manufacturer’s instruction. The isolated RNA was constructed into a fragment library using TruSeq Stranded Total RNA LT Sample Prep Kit Gold (Illumina, San Diego, CA, USA). Constructed libraries were subjected to total transcriptome enrichment using a NovaSeq 6000 S4 Reagent Kit. The prepared transcriptome libraries were sequenced on an Illumina NovaSeq 6000 platform using the paired-end sequencing method. All procedures were performed according to the manufacturer’s instructions. Reads were aligned to GRCh38 using HISAT2. Around one hundred million reads were mapped per sample. Known genes and transcripts were assembled with StringTie based on the reference genome model. We utilized the FPKM value of known genes obtained through StringTie analysis to detect the differentially expressed lncRNA. During data preprocessing, low-quality transcripts were filtered out. Afterward, log2 transformation of FPKM+1 and quantile normalization were performed. Genes showing two-fold or more changes in expression by MAPK attenuation or LINC00941 knockdown were interpreted as significant.

### 2.6. Quantitative Reverse Transcription-Polymerase Chain Reaction (qRT-PCR)

Complementary DNA (cDNA) was synthesized using the total RNA and High-Capacity cDNA Reverse Transcription Kit (Applied Biosystems, Foster City, CA, USA). Then, the quantitative reverse transcription-polymerase chain reaction (qRT-PCR) of *LINC00941* was performed using TaqMan Gene Expression Assay, TaqMan Fast Advanced Mix, and ABI 7500 Sequence Detection System (Applied Biosystems). Expression levels were calculated by the ∆∆Ct method using *GAPDH* level as an endogenous control. Similarly, qRT-PCR of *E2F7, F3*, and *CD82* with an endogenous control *B2M* was performed using primers listed in Appendix A and the PowerUp SYBR Green Master Mix (Applied Biosystems). All procedures were performed according to the manufacturer’s instructions.

### 2.7. TCGA Open-Access Database Analysis

RNA-Seq data were downloaded from The Cancer Genome Atlas (TCGA) portal (https://portal.gdc.cancer.gov accessed on 4 October 2020) for pancreatic cancer patients whose clinical data were available. The data of 177 primary tumor tissues and four normal pancreatic tissues were analyzed. To investigate the correlation between the expression level and overall survival, we divided 177 cases into a high expression group (*n* = 86) and a low expression group (*n* = 86), excluding cases with a median value, for each gene.

### 2.8. Promoter Assay

A candidate promoter region spanning 3720 base pairs (bp) adjacently upstream of *LINC00941* (NR_040245 in Refseq (https://www.ncbi.nlm.nih.gov/refseq/ accessed on 18 March 2020)) was amplified using a KOD-FX NEO DNA Polymerase Kit (TOYOBO) with the paired primers 5′-TACGCGTGCTAGCCCTTCTGTCCAACAAATTCCTCTCC-3′ and 5′-GCAGATCTCGAGCCCCCATCCGGCTCTCAGAAGTG-3′, including a portion of vector sequence for in-fusion cloning and human genomic DNA as a template in the following condition: initial denaturation for 2 min at 94 °C, 40 cycles of reactions comprising 40 s at 98 °C, 30 s at 65 °C, and 3 min and 42 s at 68 °C, then a final extension for 2 min at 68 °C. The amplified product was purified with a High-Pure PCR Product Purification Kit (Roche) and cloned into the reporter vector, pGL3-Basic (Promega, Madison, WI, USA), with an In-Fusion HD Cloning Kit (TaKaRa, Shiga, Japan) according to the manufacturer’s instructions. The cloned vector was confirmed by DNA sequencing.

Reporter vectors harboring mutated sequences of the consensus binding sites of ELK-1 and ETS-1 transcription factors were generated by site-directed mutagenesis performed with a QuikChange Ⅱ Site-directed Mutagenesis kit (Agilent Technologies, Santa Clara, CA, USA) and the mutagenic primers 5′-GGATAAGAATACTATTTATGAGCTT**TT**TGTTTGTGAATGGC-3′ for ELK-1 and 5′-CCTTGGCTTCTACCATCC**AA**AAGTCTTCTGCCAACCCC-3′ for ETS-1, respectively, with mutations at the underlined two bases according to the manufacturer’s instructions. The constructed vectors were verified by DNA sequencing.

AsPC-1 cells were seeded in a 6-well plate at 3 × 10^4^ cells per well and cultured with RPMI1640 with 10% FBS. Subsequently, the cells were subjected to culture in the following two conditions to activate MAPK pathway. (1) Transfection with the active MAP2K vector assay: 24 h after seeding, 0.5 μg of either of the constructed reporter vectors, 0.05 μg of phRL-TK vector (Promega), and 0.5 μg of a plasmid expressing the active MAP2K1/MEK (MAP2K1D44-51/S218E/S222E) [11] or pcDNA3.1 V5/HisA (empty vector) were co-transfected using Lipofectamine 3000 (Invitrogen, Carlsbad, CA, USA) reagent according to the manufacturer’s instruction. The transfected cells were maintained for 48 h. (2) FBS starvation assay: 24 h after seeding, the medium was changed to RPMI1640 without FBS. After an additional 24 h, 0.5 μg of either cloned reporter vectors and 0.05 μg of phRL-TK vector (Promega) were co-transfected. Then, 24 h after the transfection (48 h after FBS starvation), the 10% FBS was added or not added. The transfected cells were maintained for 24 h.

The cells were then washed with PBS and lysed in Lysis buffer (Promega). A dual luciferase assay using a Dual Luciferase Assay Kit (Promega) and Centro LB960 (Berthold, Bad Wildbad, Germany) was performed according to the manufacturers’ instructions.

### 2.9. Cell Proliferation Assay with Altering LINC00941 Expression

Cells were seeded in 96-well plates at 3 × 10^3^ per well for MIA PaCa-2 and AsPC-1 or at 1 × 10^3^ per well for PCI-35. Twenty-four hours after seeding, the cells were transfected with small interfering RNAs (siRNAs) for *LINC00941* (si#1, sense: 5′-GCCUCCAUAUUCAUGAACUtt-3′, antisense: 5′-AGUUCAUGAAUAUGGAGGCtg-3′, si#2, sense: 5′-CCAUUCAGCCUUGAACAUUtt-3′, antisense: 5′-AAUGUUCAAGGCUGAAUGGtc-3′, Thermo Fisher Scientific, Waltham, MA, USA) or Silencer Select Negative Control (Thermo Fisher Scientific) at 200 nmol/L using Oligofectamine Transfection Reagent (Invitrogen) according to the manufacturer’s instructions. Then, 24 h after transfection, the expression of *LINC00941* was assayed by qRT-PCR. A colorimetric cell proliferation assay employing 0.05% MTT (Sigma) was carried out as described previously [12]. For the proliferation assay with forced expression of *LINC00941*, full-length *LINC00941* cDNA was cloned into the pcDNA6/*myc*-HisA vector (Invitrogen). The cloned vectors were verified by DNA sequencing. The cells were transfected with the cloned vector or the empty vector at 1 ng/μL using Lipofectamine3000 Transfection Reagent (Invitrogen) according to the manufacturer’s instructions. Finally, 24 h after transfection, the expression of *LINC00941* was assayed by qRT-PCR, and an MTT assay was carried out in the same way.

### 2.10. RNA-Seq Data Analysis

iDEP (http://bioinformatics.sdstate.edu/idep/ accessed on 3 December 2020) was used for hierarchical clustering [13]. Metascape (http://metascape.org/ accessed on 3 December 2020) was used for the gene set enrichment analysis [14]. A gene list for analysis was generated from transcriptome sequencing analysis with the knockdown of *LINC00941*, with a total of 3229 genes showing aberrant expression and 994 genes showing aberrant expression in common between two siRNA conditions compared with the control condition.

### 2.11. Statistical Analysis

Every experiment was conducted at least twice. Statistical analyses were performed using Microsoft Excel for Mac (ver.16.43 (20110804)) and R for Mac (R 3.5.2 GUI 1.70 El Capitan build (7612)). A Student’s *t*-test was performed for comparisons between homoscedastic and parametric two groups. A Welch’s *t*-test was performed for comparisons between non-homoscedastic and parametric two groups. Kaplan–Meier survival curves were drawn, and a Log-rank test was performed to compare overall survivals between two groups. Spearman’s rank correlation test was used to determine the significance of two nonparametric variables, and *p* values less than 0.05 were considered statistically significant.

### 2.12. Data Availability

The RNA-Seq data discussed in this study have been deposited in NCBI’s Gene Expression Omnibus and are accessible through GEO Series accession numbers GSE216708 (https://www.ncbi.nlm.nih.gov/geo/query/acc.cgi?acc=GSE216708 accessed on 18 March 2020) and GSE216710 (https://www.ncbi.nlm.nih.gov/geo/query/acc.cgi?acc=GSE216710 accessed on 18 March 2020).

## 3. Results

### 3.1. LncRNAs Associated with MAPK

To uncover lncRNAs modulated by the MAPK/ERK pathway in pancreatic cancer, we carried out comparative transcriptome sequencing assays using the pancreatic cancer cell lines MIA PaCa-2, AsPC-1, and PCI-35 with or without attenuation of MAPK by U0126, a MAP2K inhibitor (Figure 1A). The transcriptome sequencing revealed that 2383 lncRNAs were evidently expressed in the cell lines in total (Appendix A). Among them, 45 lncRNAs revealed two-fold or more changes in expression level upon the attenuation of MAPK in at least one of the three cell lines to the level of statistical significance (Appendix A). Among these, eight lncRNAs, namely, *CCDC18-AS1, LINC00941, LINC-PINT, NEAT1, PRR7-AS1, SNAI3-AS1, SNHG4,* and *SPRY4-IT1,* revealed two-fold or more changes in two or three cell lines. Then, *LINC00941* was found to be significantly down-regulated in all the three cell lines (Figure 1B). The downregulation of *LINC00941* by MAPK attenuation was validated by qRT-PCR (Figure 1C). The expression of *LINC00941* was higher in AsPC-1 than in MIA PaCa-2 or PCI-35 despite the modest level of phosphorylated ERK, which suggests that the expression of *LINC00941* might depend only on both MAPK activity and on other molecules.

### 3.2. Prognostic Impacts of lncRNAs Associated with MAPK

We examined the expression of lncRNAs uncovered to be associated with MAPK activity in pancreatic cancer tissues using the TCGA portal open-access database. We focused on the eight lncRNAs showing aberrant expression in two or three cell lines, in which *LINC00941* was found to be expressed significantly higher in tumor tissues than in normal tissues (Table 1, Figure 1D). In an analysis of the association with survival, the expression levels of *LINC00941*, *LINC-PINT*, *CCDC18-AS1*, and *SNAI3-AS1* were significantly correlated with overall survival of patients. *LINC00941* indicated poorer prognosis at higher expression, while the others indicated poorer prognosis at lower expression (Table 1, Figure 1E). 

### 3.3. LINC00941 Promoter Activity Associated with MAPK

The transcriptome sequencing analysis suggested that MAPK may positively regulate *LINC00941*. To determine whether *LINC00941* is directly promoted by MAPK, we performed a promoter assay. We hypothesized that an upstream CpG-rich region of *LINC00941* would play a promoter role; hence, we cloned the upstream region spanning 3720 bp of *LINC00941* into a reporter vector (Figure 2A). The reporter vector was transfected into the pancreatic cancer cell lines. Then, the activity of MAPK was modulated by the transfection of the constitutively active MAP2K vector or FBS starvation. We found that the upstream region had promoter activity tightly concordant with MAPK activity (Figure 2B).

Next, we searched for possible binding sites of transcription factors associated with MAPK in the active promoter region. With help from a web-based searching program for transcription factor-binding sites, Match (http://www.gene-regulation.com/cgi-bin/pub/programs/match/bin/match.cgi accessed on 18 March 2020), we found consensus binding sequences of the ELK-1 transcription factor CTTCC and ETS-1 transcription factor CGGAA in the region. We then constructed reporter vectors containing mutated sequences of the consensus binding site, i.e., CTTCC and CGGAA into CTTTT and CAAAA, respectively, and analyzed their promoter activities (Figure 2A). While the ETS-1 mutant vector showed significantly less activity, it did show a certain promoter activity compared to the wild-type sequence vector, while the ELK-1 mutant vector showed no significant alteration, indicating that ETS-1 is one of transcription factors significantly associated with the promoter activity of *LINC00941* (Figure 2C).

### 3.4. Effect of LINC00941 on Cell Proliferation

To determine the functional significance of *LINC00941* in pancreatic cancer, we examined whether cellular proliferation is altered by altering the expression of *LINC00941* in cultured pancreatic cancer cells. Transfection of siRNAs targeting *LINC00941* induced significant suppression of endogenous expression of *LINC00941* in all examined cell lines (Figure 3A). In vitro proliferation assays showed that proliferation of MIA PaCa-2 and AsPC-1 were inhibited by the suppression of *LINC00941*, while that of PCI-35 was not changed (Figure 3B–D). On the other hand, transfection of the pcDNA vector harboring the full-length *LINC00941* cDNA induced overexpression of *LINC00941* and upregulation of cell proliferation in all of the examined cell lines (Figure 3E–H). The inconsistent results in PCI-35 might be due to feedback effects involving unknown molecules.

### 3.5. Target Genes of LINC00941

To investigate downstream targets of *LINC00941*, we performed a comparative transcriptome sequencing analysis of AsPC-1 with or without *LINC00941* knockdown, because AsPC-1 showed the most robust endogenous expression of *LINC00941* with the largest difference in *LINC00941* expression by siRNA-mediated knockdown. We analyzed genes differentially expressed between a negative control (non-specific siRNA) and the knockdown of *LINC00941* with si#1 or si#2. The scheme of processing data obtained by total transcriptome sequencing is drawn in Figure 4A. In 16,463 genes expressed in AsPC-1, we found that 3229 genes showed two-fold or more changes with modulation of *LINC00941*. A hierarchical clustering heatmap suggests the similarity between two siRNA conditions (Figure 4B). Among them, 416 and 578 genes were significantly up-regulated and down-regulated in common between the two siRNA conditions, respectively, compared with the negative control (Appendix A). Metascape enrichment analysis using GO BP terms showed that the genes modulated by *LINC00941* were associated with 44 biological processes, including glycoprotein biosynthetic process, beta-catenin-TCF complex assembly, and histone H3-K4 monomethylation. The top twenty GO BP terms are shown in Figure 4C and Appendix A.

We compared the data from the transcriptome analyses of MAPK-attenuated and *LINC00941*-attenuated in AsPC-1. We found that six genes showed concordant alterations in expression in the two conditions: Downregulated genes were *E2F7* and *SPRY4-IT1*, while upregulated genes were *CD82*, *F3*, *LOC100507291*, and *TNNC1* (Table 2). TCGA database analysis revealed that *E2F7, CD82*, and *TNNC1* showed significantly higher expression in tumor tissues than in normal tissues (Table 2, Figure 4D). In addition, higher expression of *E2F7, F3*, and *CD82* were associated with poor prognosis (Table 2, Figure 4E). In addition, *E2F7, F3*, and *CD82* showed higher expression levels than *SPRY4-IT1, LOC100507291*, and *TNNC1* (Appendix A). We focused on *E2F7, F3*, and *CD82* and validated the transcriptome sequencing result by qRT-PCR. *E2F7* showed tightly allied expression with *LINC00941* expression, indicating that *LINC00941* directly regulates *E2F7* (Figure 4F,G). On the other hand, *CD82* and *F3* were upregulated in both knockdown and forced expression of *LINC00941*, suggesting that *LINC00941* may not directly regulate them (Appendix A). The expression levels of *LINC00941* and *E2F7* were positively correlated in the TCGA data (Spearman’s rank correlation rho = 0.57, *p* < 2.2 × 10^−16^; Appendix A). When the patients were divided into four groups according to levels of expression of *LINC00941* and *E2F7*, namely, low-*LINC00941* and low-*E2F7* (low/low), low-*LINC00941* and high-*E2F7* (low/high), high-*LINC00941* and low-*E2F7* (high/low), and high-*LINC00941* and high-*E2F7* (high/high), the overall survival was more favorable in the low/low group than the rest of groups, which suggests that low expression of both *LINC00941* and *E2F7* can be a biomarker predicting the favorable prognosis (*LINC00941* level/*E2F7* level, Log-rank test: low/low vs. low/high, *p* = 0.04; low/low vs. high/low, *p* = 3 × 10^−4^; low/low vs. high/high, *p* = 0.002, Appendix A).

## 4. Discussions

This study identified lncRNAs associated with MAPK activity in pancreatic cancer cells. Transcriptome sequencing showed that 2383 lncRNAs were evidently expressed; among these, 45 lncRNAs were found to be significantly associated with MAPK activity. We determined that *LINC00941* was commonly promoted by MAPK in the examined pancreatic cancer cell lines. In TCGA data analysis, the high expression of *LINC00941* was associated with poor prognosis of patients with pancreatic cancer. The in vitro proliferation analysis showed that *LINC00941* promoted the proliferation of pancreatic cancer cells. Moreover, we identified that the 5′-upstream region spanning 3700-bp adjacent to exon 1 of *LINC00941* had promoter activity, which was indeed associated with MAPK activity. ETS-1, the downstream target gene of MAPK, was suggested to be one of the transcription factors involved in the expression of *LINC00941*. Furthermore, we identified potential target genes of *LINC00941* associated with various biological processes. These results indicate that MAPK regulates lncRNAs, and that *LINC00941* is one such regulated lncRNA associated with the poor prognosis of pancreatic cancer.

LncRNAs are a group of non-protein coding RNA molecules greater than 200 nucleotides [6], and have been proven to play important roles in cancer pathogenesis. LncRNAs function as guides of transcription factors, scaffolds of multiple molecules, and decoys of miRNA intervening interactions with target genes [6,7]. In cancers, lncRNAs are supposed to act as oncogenes and/or tumor suppressor genes, which may contribute to the development and progression of cancers [6,7]. There is great interest in identifying lncRNAs applicable to cancer biomarkers and/or therapeutic targets [15]. However, how lncRNAs are aberrantly expressed in cancer cells is largely unknown. Cancer-associated aberrant signaling pathways play a central role in the development and progression of cancer [16]. These aberrant signals modulate the expression of genes and alter the fate of cells. LncRNAs in cancers might be regulated by aberrant signaling pathways as well; therefore, we explored lncRNAs associated with MAPK in pancreatic cancer. Pancreatic cancer is characterized by constitutive activation of the MAPK pathway. Mutations of *KRAS* or *BRAF* and epigenetic abrogation of DUSP6 contribute synergistically to the constitutive activation of MAPK [3]. Active MAPK induces the expression of various genes that are thought to play roles in malignant phenotypes of pancreatic cancer. The downstream genes of MAPK are involved in the cell cycle and mitosis, DNA replication, receptor signaling, and transport. The downstream target genes of MAPK are not only protein-coding genes, they are noncoding RNA genes. We previously identified miRNAs preferentially associated with MAPK, namely, miR-7-3, miR-34a, miR-181d, and miR-193b, in which miR-7-3 is the oncogenic miRNAs, whereas miR-34a, miR-181d, and miR-193b are tumor-suppressive miRNAs. These miRNAs are useful as serum biomarkers to discriminate pancreatic cancer from benign diseases [8,9].

We uncovered 45 lncRNAs significantly associated with MAPK activity. Among them, *LINC00941* was consistently promoted by MAPK in the examined pancreatic cancer cell lines. *LINC00941* was associated with cell proliferation. Moreover, TCGA data analysis indicated that *LINC00941* was found to be upregulated in pancreatic cancer tissues and associated with poor survival of patients. Recent studies have demonstrated that *LINC00941* is involved in tumorigenesis and malignant phenotypes of several kinds of cancers, including pancreatic cancer. In pancreatic cancer, it has been reported that that *LINC00941* promotes cancer cell growth by enhancing aerobic glycolysis through modulation of the Hippo pathway [17]. Another study has demonstrated that *LINC00941* is associated with epithelial-mesenchymal transition (EMT), acting as a competitive endogenous RNA for miR-335-5p [18]. In other cancers, *LINC00941* is often highly expressed in tumor tissues compared to normal tissues. This higher expression is associated with poorer prognosis in hepatocellular carcinoma, lung adenocarcinoma, papillary thyroid carcinoma, gastric cancer, colorectal cancer, and oral squamous cell carcinoma [19,20,21,22,23,24,25]. Several studies in other cancers have suggested that *LINC00941* promotes the epithelial-to-mesenchymal transition (EMT) [18,19,23,24]. Yan et al. demonstrated that *lncRNA-MUF/LINC00941* activated EMT in hepatocellular carcinoma cells by interaction with the Annexin A2 (ANXA2) and miR-34a [19]. Wu et al. demonstrated that *LINC00941* activated EMT in colorectal cancer cells via activating the TGF-β/SMAD2/3 axis by directly binding the SMAD4 protein [23]. Furthermore, Ai et al. demonstrated that *LINC00941* promoted the growth of oral squamous cell carcinoma cells. They showed that EP300 activated *LINC00941* transcription and *LINC00941* promoted Wnt/β-catenin signaling with CAPRIN2 activation [24]. Therefore, *LINC00941* is indicated as one of the major onco-lncRNAs in various cancers, including pancreatic cancer. However, how the expression of *LINC00941* is modulated in cancer tissues has been unknown. In this study, we prove that the expression of *LINC00941* is tightly associated with MAPK activity in pancreatic cancer cells.

We identified the promoter region of *LINC00941*, which was actually associated with MAPK activity via ETS-1, not ELK-1, consensus binding site. ETS-1 is a member of the ETS transcription factor family, which is known to be a critical player in cancer pathogenesis as a mediator of MAPK signaling activity [26,27,28,29]. The relationship between the ETS family and lncRNAs has been proven by several studies, and it has been reported that ELK-1 induces the upregulation of lncRNAs and is associated with the growth of cancer cells [30,31]. On the other hand, lncRNAs are reported to be involved in regulating the ETS family [32,33]. Therefore, ETS-family molecules and lncRNAs seem to make up a feedback network, and our results consistently indicate that *LINC00941* can be modulated by ETS-1.

*E2F7* is likely to be one of the downstream targets of *LINC00941*. *E2F7* encodes transcription factor E2F7, which has been reported to be essential in regulating cell cycle progression [34]. *E2F7* has been reported to be highly expressed in pancreatic cancer tissues, and higher expression of *E2F7* was associated with the poorer prognosis [35,36]. Lu et al. suggested that lncRNA *CASC19* contributes to the progression of pancreatic cancer by modulating miR148b/*E2F7* axis [37]. Xu and Qi suggested that miR-10b inhibits invasion by migration of pancreatic cancer cells by regulating *E2F7* expression, and that the high expression of *E2F7* is associated with poor prognosis [38]. These previous studies and our study show that various ncRNAs are associated with regulating *E2F7*, which may play an important role in the malignant phenotype of pancreatic cancer.

The 45 lncRNAs associated with MAPK activity are likely to be implicated in the pathogenesis of pancreatic cancer. Indeed, *BLACAT1, CASC8, HCP5, LINC-PINT, SNHG1, SNHG9, SNHG12*, and *SPRY4-IT1* were detected as MAPK-promoted lncRNAs, and *NEAT1, PSMB8-AS1, THAP9-AS1*, and *TP53TG1* were detected as MAPK-inhibited lncRNAs reported to be involved in pancreatic cancer [39,40,41,42,43,44,45,46,47,48,49,50,51]. For example, previous studies reported that *LINC-PINT* levels in plasma are lower in pancreatic cancer patients than healthy controls [42,51], and that *LINC-PINT* suppresses the growth of pancreatic cancer cells through TGF-β pathway activation [51], which suggests that *LINC-PINT* can be a diagnostic and a prognostic biomarker of pancreatic cancer. *NEAT1* was reported to be highly expressed in pancreatic cancer tissues, and the knockdown of *NEAT1* suppressed the growth of pancreatic cancer cells [32]. Interestingly, the activation of MAPK is considered to play a role in promoting malignant phenotypes of pancreatic cancer [3]. However, the MAPK-promoted lncRNAs include both those reported as oncogenic and tumor-suppressive and vice versa for the MAPK-inhibited lncRNAs. Several of these lncRNAs might have an unappreciated function or show a conflicting result of a feedback loop of MAPK signaling pathway in pancreatic cancer. Nevertheless, these MAPK-associated lncRNAs could be therapeutic targets as well as diagnostic biomarkers, as already shown in *LINC-PINT* and *NEAT1* [32,51]. Further research is needed in order to understand the functions of lncRNAs associated with MAPK.

## Figures and Tables

**Figure 1 cancers-15-00303-f001:**
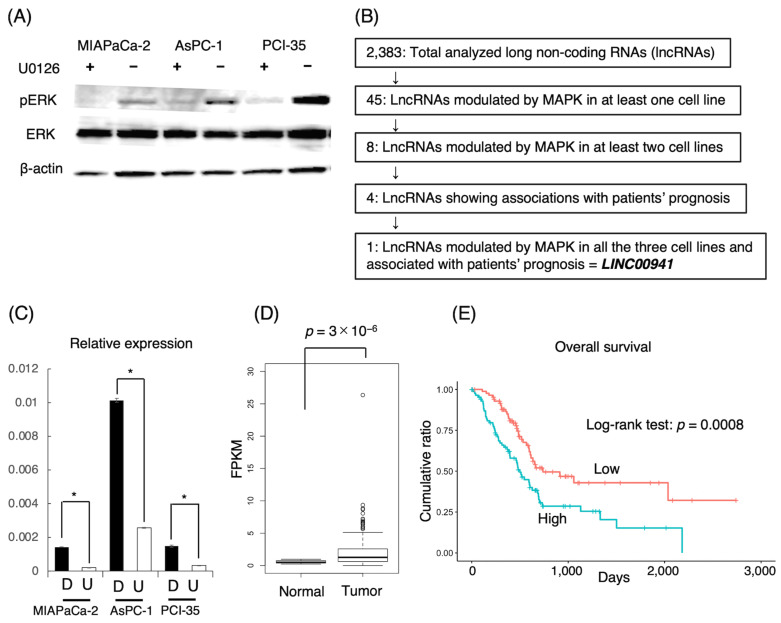
MAPK-associated lncRNAs identified by a comparative transcriptome sequencing assay of pancreatic cancer cells with or without attenuation of MAPK activity. (**A**) The expression of phosphorylated ERK was inhibited by U0126, a MAP2K inhibitor, in pancreatic cancer cell lines. (**B**) Stringent processing of transcriptome sequencing data resulted in identification of MAPK-associated lncRNAs. (**C**) qRT-PCR assay validated the transcriptome data revealing downregulation of *LINC00941* by the attenuation of MAPK in pancreatic cancer cell lines (* *p* < 0.05). Abbreviations: D, DMSO; U, U0126. (**D**) TCGA data analysis showed that *LINC00941* was expressed significantly more highly in pancreatic cancer tissues than in normal tissues (*p* = 3 × 10^−6^). (**E**) TCGA data analysis showed that highly expressed *LINC00941* in pancreatic cancer tissues was associated with patients’ poor overall survival (*p* = 0.0008).

**Figure 2 cancers-15-00303-f002:**
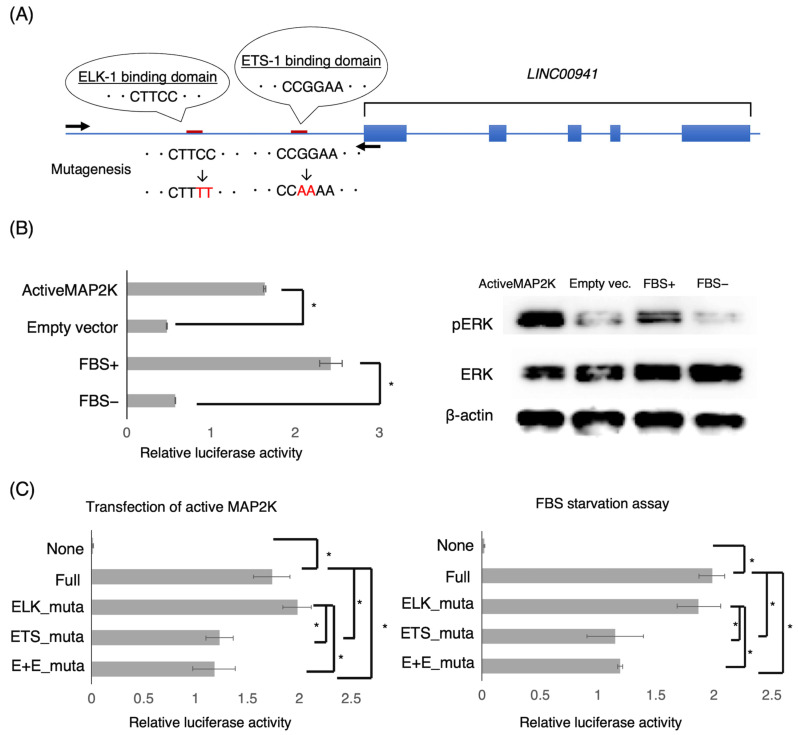
*LINC00941* promoter activity is associated with MAPK activity. (**A**) A candidate promoter region of *LINC00941* harboring consensus sequences of biding sites for ELK-1 and ETS-1. Arrows indicate relative positions of primers spanning the 3720-bp genomic region immediately upstream of exon 1 of *LINC00941* used for construction of reporter vectors for the promoter assay. Mutagenic disruptions of the consensus binding sites used in experiments shown in (**C**) are indicated. (**B**) Reporter assays with modulation of MAPK activity. Left panel: the candidate promoter region reveals the promoter activity significantly associated with MAPK activity modulated by the active MEK/MAP2K transfection or FBS starvation in AsPC-1 (* *p* < 0.05). Right panel: Immunoblots showing upregulation and downregulation of MAPK activity in AsPC-1 by the active MEK transfection and FBS starvation, respectively. (**C**) Mutagenesis of the consensus binding sites in the promoter of *LINC00941* results in a significant reduction of promoter activity by disruption of the binding site of ETS-1, and not that of ELK-1, in both active MEK transfection and FBS starvation in AsPC-1 (* *p* < 0.05). E+E means disruption of both binding sites of ETS-1 and ELK-1.

**Figure 3 cancers-15-00303-f003:**
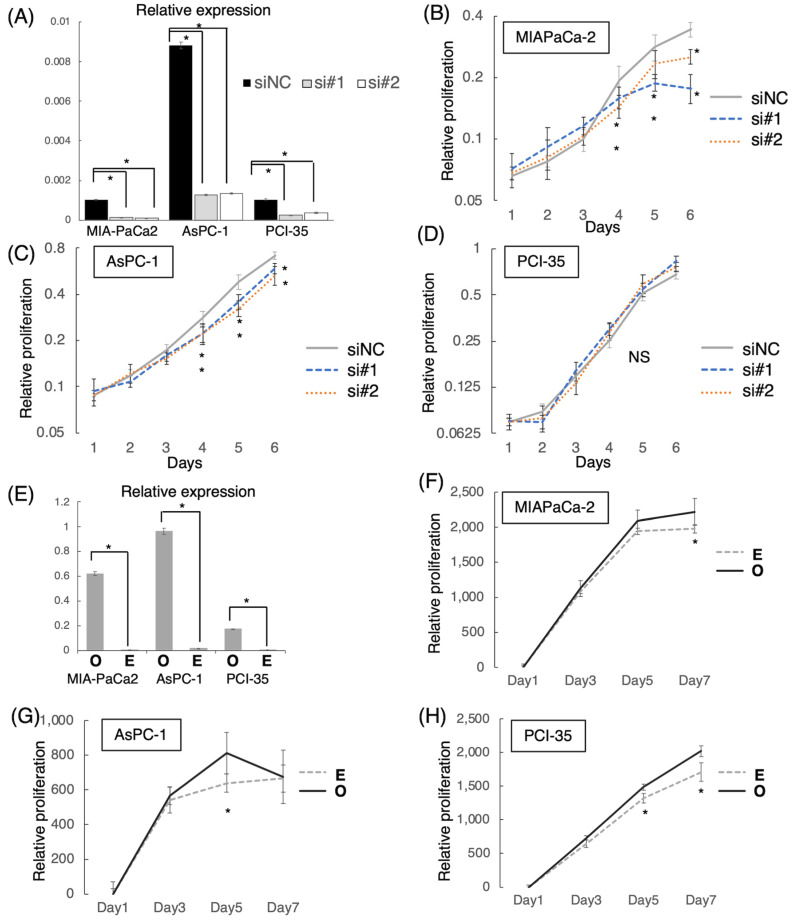
MTT proliferation assays with the knockdown or overexpression of *LINC00941*. (**A**) qRT-PCR validation of downregulation of *LINC00941* expression by siRNAs. NC, negative control. (**B**–**D**) The knockdown of *LINC00941* suppressed cell proliferation of MIAPaCa-2 and AsPC-1 (* *p* < 0.05), and did not suppress PCI-35. (**E**) qRT-PCR validation of the overexpression of *LINC00941* by cDNA vector transfection. Abbreviations: E, empty vector; O, overexpression. (**F**–**H**) The overexpression of *LINC00941* promoted the cell proliferation of MIAPaCa-2, AsPC-1, and PCI-35. Please note that the values of negative controls of experiments in (**A**,**E**) were comparable in each cell line: 0.0010 and 0.0015, 0.0088 and 0.017, and 0.0010 and 0.0019 for siNC and E in MIA PaCa-2, AsPC-1, and PCI-35, respectively.

**Figure 4 cancers-15-00303-f004:**
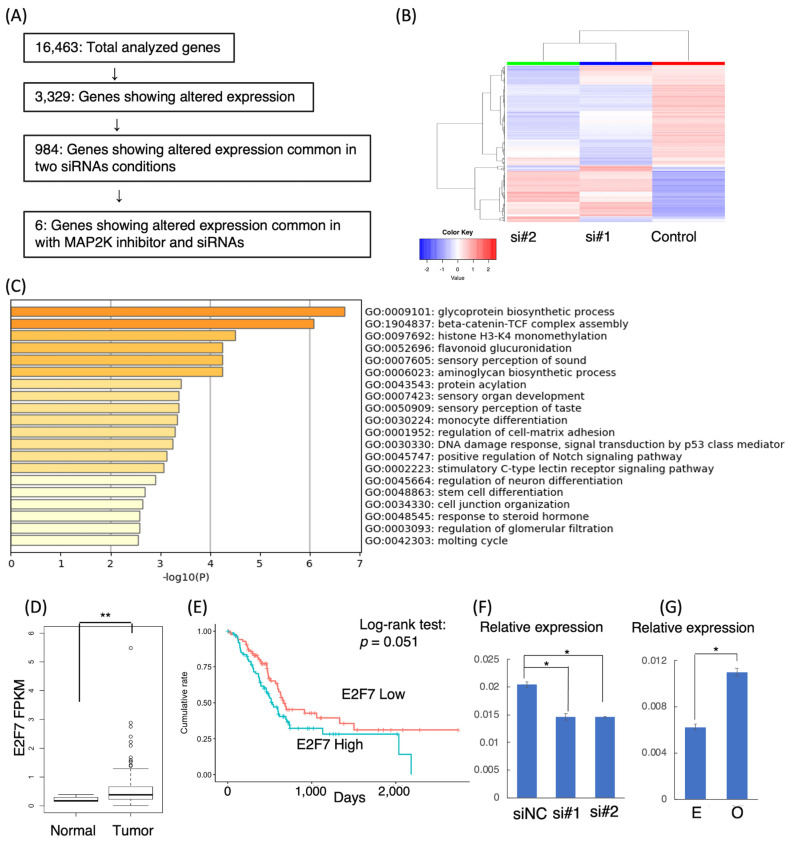
Identification of genes associated with *LINC00941.* (**A**) Stringent processing of comparative transcriptome sequencing data of AsPC-1 with or without attenuation of the expression of *LINC00941*. (**B**) Heatmap with hierarchical clustering showing comparison of transcriptome data between cells with siRNAs for *LINC00941* and negative control. (**C**) Top 20 Gene Ontology Biological Process terms of genes associated with *LINC00941.* (**D**) TCGA data analysis showed that pancreatic cancer tissues revealed higher expression of *E2F7* than normal tissues (** *p* < 0.001). (**E**) TCGA data analysis showed the marginal survival difference between patients with pancreatic cancer with higher and lower expression of *E2F7* (*p* = 0.051 by log-rank test). (**F**) qRT-PCR validated the transcriptome data showing downregulation of *E2F7* by knockdown of *LINC00941* in AsPC-1 (* *p* < 0.05). (**G**) qRT-PCR showed that overexpression of *LINC00941* in AsPC-1 resulted in upregulation of *E2F7* (* *p* < 0.05). Abbreviations: E, empty vector; O, overexpression.

**Table 1 cancers-15-00303-t001:** MAPK-regulated lncRNAs.

Gene Symbol	Transcript ID	Effect of MAPK	*p*-Value for the Difference of Expression in Tumor (T) vs. Normal (N) in TCGA	*p*-Value (Log-Rank) for Overall Survival in TCGA	Expression Level Associated with Poor Prognosis
*CCDC18-AS1*	NR_034089	Downregulation	0.61 (N > T)	0.009	Low
*LINC00941*	NR_040245	Upregulation	3 × 10^−6^ (T > N)	0.0008	High
*LINC-PINT*	NR_015431	Upregulation	0.55 (N > T)	0.01	Low
*NEAT1*	NR_028272	Downregulation	0.86 (N > T)	0.8	-
*PRR7-AS1*	NR_038915	Upregulation	0.65 (N > T)	0.07	-
*SNAI3* *-AS1*	NR_024399	Downregulation	0.29 (N > T)	0.00001	Low
*SNHG4*	NR_003141	Upregulation	0.99 (T > N)	0.7	-
*SPRY4-IT1*	NR_131221	Upregulation	0.22 (T > N)	0.2	-

**Table 2 cancers-15-00303-t002:** Concordantly altered genes between MAPK attenuation and knockdown of *LINC00941*.

Gene Symbol	Transcript ID	Effect of MAPK and *LINC00941*	*p*-Value for the Difference of Expression in Tumor (T) vs. Normal (N) in TCGA	*p*-Value (Log-Rank) for Overall Survivalin TCGA	Expression Level Associated with Poor Prognosis
*CD82*	NM_001024844	Downregulation	0.01 (T > N)	0.04	High
*E2F7*	NM_203394	Upregulation	0.001 (T > N)	0.05	High
*F3*	NM_001178096	Downregulation	0.36 (T > N)	0.0001	High
*LOC100507291*	NR_121608	Downregulation	0.10 (T > N)	0.3	-
*SPRY4-IT1*	NR_131221	Upregulation	0.22 (T > N)	0.2	-
*TNNC1*	NM_003280	Downregulation	7 × 10^−6^ (T > N)	0.5	-

## Data Availability

The RNA-Seq data discussed in this study have been deposited in NCBI’s Gene Expression Omnibus and are accessible through GEO Series accession numbers GSE216708 (https://www.ncbi.nlm.nih.gov/geo/query/acc.cgi?acc=GSE216708 accessed on 18 March 2020) and GSE216710 (https://www.ncbi.nlm.nih.gov/geo/query/acc.cgi?acc=GSE216710 accessed on 18 March 2020).

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
