# Peer review of "Long Non-Coding RNAs Associated with Mitogen-Activated Protein Kinase in Human Pancreatic Cancer"

_cancers, 2023, doi:10.3390/cancers15010303_

Round 1

Reviewer 1 Report

The manuscript reports an RNA-seq analysis of 3 pancreatic cancer cell lines cultured with or without U0126 (a MAP2K inhibitor) to evaluate the role of the MAPK pathway in lncRNA. It also explores the prognostic relevance of lncRNAs identified using the TCGA database focusing on LINC00941. The authors also studied the promoter activity of MAPK-regulated genes and identified ETS-1 in a promoter domain of LINC00941. In vitro experiments revealed that LINC00941 accelerates cell proliferation. Further TCGA database analysis showed that expression of the E2F7 gene was correlated with LINC00941.

This is an interesting and innovative study. I have several comments for improvement.

1.     The authors showed the correlation between E2F7 and lncRNA LINC00941.  It would be interesting to evaluate whether it correlates with patient outcomes.

2.     Line 32-33 – the authors mention that this is "valid as therapeutic targets as well as diagnostic biomarkers". But in the Discussion, there is no specific reference to its uses for diagnosis or therapy.  

3.     Line 227 - since the authors mentioned "MAPK signaling", did they also look into other MAPK pathways, such as p38 and JNK?

4.     Fig 1B - in Line 245, 4 genes (LINC00941, LINC-PINT, CCDC18-AS1, SNAI3-AS1) were sub-grouped from the 8 genes that were relevant in at least two cell lines.  It might be easier to follow if they are added to the flow chart.

5.     Fig 1C - qPCR data are shown as delta CT values only.  Why not 2^-deltaCT values?

6.     Fig 1C - Why was the AsPC-1 DMSO value higher than the other cell lines?

7.     Fig 1C and 1A - according to WB, MAPK pathway looks like PCI-35 > AsPC-1 > MIAPaCa-2, but at RNA level it looks like AsPC-1 >> MIAPaCa-2 = PCI-35. How could we reconcile this discrepancy?

8.     Table 1,2 – it would be easier to follow if listed in alphabetical order, Transcript ID order, p-value order, or the prognosis association order.

9.     Fig 2B - Y axis label missing.

10.  Fig 2B & 2C - Why are ETS_muta and E+E_muta elevated compared to Empty vector? 

11.  Fig 3D - Why was PCI-35 growth curve not significant despite LINC00941 knock down?

12.  Fig 3E - Why is the deltaCt value inconsistent with the previous control values for empty vectors?

13.  Fig 3G - it looks like the day1 graph has a minus quartile value. Why? Fig F, G, H - there is no need to start the range of the Y-axis from minus.

14.  Fig 3 B-H: What are the units? 

15.  Line 283 - knockdown AsPC-1 cells expressed relatively high levels of LINC00941 (comparable with MApaCa-2, PCI-35 normal cells). Was it valid to use those cells?

16.  Fig 4C - since Metascape enrichment analysis using GO BP terms were performed, it would be nicer if the genes related to LINC00941 were referred to which biological process it was related to.

17.  Line 299, 301, 303, 304 - CD-82, CD82, CD83(2?), CD83(2?) need re-checking.

18.  Line 315 "poo" -> "poor"

19.  Line 321 - "various biological process" ... only proliferation in vitro(?)

20.  Line 323 - "which contributes to poor prognosis" ... not shown

21.  Table 1,2 - some "P-value for Tumor vs. Normal" lanes have "(T > N)" and some don't.

22.  Supplementary Table 1 - is the annealing temperature 60 â„ƒ fr all primers?

23.  The supplementary excel file tab name is "Sapplementary Table 4"

24.  Supplementary Table 4, Table 5 - title "folod" -> "fold"

25.  Supplementary Fig - down regulation has a space, upregulation doesn't

26.  Some descriptions of E2F7 are not Italic, need to check if referring to protein level or DNA/RNAs

27.  S-Fig. 1: Are the upregulations of F3 and CD82 in knockdown and overexpression of LINC00941 significant as well? If yes - how would they explain it?

English needs to be checked thoroughly.

Author Response

Reviewer #1:

Comments and Suggestions for Authors

The manuscript reports an RNA-seq analysis of 3 pancreatic cancer cell lines cultured with or without U0126 (a MAP2K inhibitor) to evaluate the role of the MAPK pathway in lncRNA. It also explores the prognostic relevance of lncRNAs identified using the TCGA database focusing on LINC00941. The authors also studied the promoter activity of MAPK-regulated genes and identified ETS-1 in a promoter domain of LINC00941. In vitro experiments revealed that LINC00941 accelerates cell proliferation. Further TCGA database analysis showed that expression of the E2F7 gene was correlated with LINC00941.

This is an interesting and innovative study. I have several comments for improvement.

  1. The authors showed the correlation between E2F7 and lncRNA LINC00941.  It would be interesting to evaluate whether it correlates with patient outcomes.

Response: Thank you for your thoughtful suggestion. We already showed the impact of E2F7 expression on patients’ survivals in Figure 4E. Then, we performed an additional TCGA data analysis to explore whether the correlation between E2F7 and LNC00941 was associated with survivals. The result of this additional analysis is shown in supplementary figure S3 and described at L.320-329 as follows: The expression levels of LINC00941 and E2F7 were positively correlated in TCGA data (Spearman’s rank correlation rho= 0.57, p < 2.2×10-16; Supplementary Figure S3). When the patients were divided into four groups according to levels of expression of LINC00941 and E2F7, namely, low-LINC00941 and low-E2F7 (low/low), low-LINC00941 and high-E2F7 (low/high), high-LINC00941 and low-E2F7 (high/low), and high-LINC00941 and high-E2F7 (high/high), the overall survival was most favorable in the low/low group than the rest of groups, which suggests that the low expression of both LINC00941 and E2F7 can be a biomarker predicting the favorable prognosis (LINC00941 level/E2F7 level, Log-rank test: low/low vs. low/high, p = 0.04; low/low vs. high/low, p = 3×10-4; low/low vs. high/high, p = 0.002, Supplementary Figure S3).

  1. Line 32-33 – the authors mention that this is "valid as therapeutic targets as well as diagnostic biomarkers". But in the Discussion, there is no specific reference to its uses for diagnosis or therapy.  

Response: We added a description in the discussion at L. 429-431 as follows: Nevertheless, these MAPK-associated lncRNAs could be therapeutic targets as well as diagnostic biomarkers as already shown in LINC-PINT and NEAT1 [32, 51].

  1. Line 227 - since the authors mentioned "MAPK signaling", did they also look into other MAPK pathways, such as p38 and JNK?

Response: In this study, we focused on MAPK/ERK pathway; therefore, we revised the text as follows: Line 17, the significance of mitogen-activated protein kinase/extracellular signal-regulated kinase (MAPK/ERK). Line 44, the mitogen-activated protein kinase/extracellular signal-regulated kinase (MAPK/ERK). Line 284, MAPK/ERK pathway

  1. Fig 1B - in Line 245, 4 genes (LINC00941, LINC-PINT, CCDC18-AS1, SNAI3-AS1) were sub-grouped from the 8 genes that were relevant in at least two cell lines.  It might be easier to follow if they are added to the flow chart.

Response: Fig 1B was revised according to this suggestion.

  1. Fig 1C - qPCR data are shown as delta CT values only.  Why not 2^-deltaCT values?

Response: We used ∆∆Ct method. We revised the method at line 134 as follows: Expression levels were calculated by ∆∆Ct method using GAPDH level as an endogenous control.

  1. Fig 1C - Why was the AsPC-1 DMSO value higher than the other cell lines?
  2. Fig 1C and 1A - according to WB, MAPK pathway looks like PCI-35 > AsPC-1 > MIAPaCa-2, but at RNA level it looks like AsPC-1 >> MIAPaCa-2 = PCI-35. How could we reconcile this discrepancy?

Response: Regarding these comments (#6 and 7), we revised the text at line 241 as follows: The expression of LINC00941 was higher in AsPC-1 than that in MIA PaCa-2 or PCI-35 despite the modest level of phosphorylated ERK, which suggests that the expression of LINC00941 might depend not only on the MAPK activity but also on other molecules.

  1. Table 1,2 – it would be easier to follow if listed in alphabetical order, Transcript ID order, p-value order, or the prognosis association order.

Response: We revised tables according to this comment.

  1. Fig 2B - Y axis label missing.

Response: We corrected figures by adding “Relative luciferase activity” as the label.

  1. Fig 2B & 2C - Why are ETS_muta and E+E_muta elevated compared to Empty vector? 

Response: According to this comment, we revised the text at line 274 as follows: The ETS-1 mutant vector showed significantly less but a certain promoter activity than the wild-type sequence vector, while the ELK-1 mutant vector showed no significant alteration, which indicated that ETS-1 is one of transcription factors significantly associated with the promoter activity of LINC00941 (Fig. 2C).

  1. Fig 3D - Why was PCI-35 growth curve not significant despite LINC00941 knock down?

Response: According to this comment, we added the following description at line 287; The inconsistent results in PCI-35 might be due to feedback effects involving unknown molecules.

  1. Fig 3E - Why is the deltaCt value inconsistent with the previous control values for empty vectors?

Response: Actually, the values of control are comparable between Fig. 3A and 3E. We added the following description in legend for Figure 3 as follows: Please note that the values of negative controls of experiments in (A) and (E) were comparable in each cell line: 0.0010 and 0.0015, 0.0088 and 0.017, 0.0010 and 0.0019 for siNC and E in MIA PaCa-2, AsPC-1, and PCI-35, respectively.

  1. Fig 3G - it looks like the day1 graph has a minus quartile value. Why? Fig F, G, H - there is no need to start the range of the Y-axis from minus.

Response: According to this comment, we revised Fig 3F,G,and H. 

  1. Fig 3 B-H: What are the units? 

Response: We added a Y-axis label, “Relative proliferation”.

  1. Line 283 – knockdown AsPC-1 cells expressed relatively high levels of LINC00941 (comparable with MapaCa-2, PCI-35 normal cells). Was it valid to use those cells?

Response: We thought that the high level of endogenous expression was most likely to give significant impact for global transcription profile in the cell. To respond to this comment, we revised the text at line 293 as follows: AsPC-1 showed the most robust endogenous expression of LINC00941 with the largest difference in LINC00941 expression by the siRNA-mediated knockdown.

  1. Fig 4C – since Metascape enrichment analysis using GO BP terms were performed, it would be nicer if the genes related to LINC00941 were referred to which biological process it was related to.

Response: According to this comment, we added Supplementary Table 6 that showed BP terms and corresponding genes.

  1. Line 299, 301, 303, 304 – CD-82, CD82, CD83(2?), CD83(2?) need re-checking.

Response: We corrected them to CD82.

  1. Line 315 "poo" -> "poor"

Response: We corrected it.

  1. Line 321 - "various biological process" ... only proliferation in vitro(?)

Response: The description was about target genes of LINC00941. We revised the sentence at line 342 as follows: Furthermore, we identified potential target genes of LINC00941, which were associated with various biological processes.

  1. Line 323 - "which contributes to poor prognosis" ... not shown

Response: The sentence was about LINC00941 that was associated with prognosis. We revised it at line 344 as follows: LINC00941 is one of such regulated lncRNAs associated with the poor prognosis of pancreatic cancer.

  1. Table 1,2 - some "P-value for Tumor vs. Normal" lanes have "(T > N)" and some don't.

Response: We revised tables to show differences between T and N.

  1. Supplementary Table 1 - is the annealing temperature 60 â„ƒ fr all primers?

Response: Yes, it is. We added estimated melting temperatures in supplementary table 1.

  1. The supplementary excel file tab name is "Sapplementary Table 4"

Response: We corrected it.

  1. Supplementary Table 4, Table 5 - title "folod" -> "fold"

Response: We corrected it.

  1. Supplementary Fig - down regulation has a space, upregulation doesn't

Response: We deleted the space between down and regulation.

  1. Some descriptions of E2F7 are not Italic, need to check if referring to protein level or DNA/RNAs

Response: We corrected some of them and confirmed that the italic is precisely used for genes.

  1. S-Fig. 1: Are the upregulations of F3 and CD82 in knockdown and overexpression of LINC00941 significant as well? If yes - how would they explain it?

Response: We added statistical significance in S-Fig.1 and additionally described at L.318 as below: On the other hand, CD82 and F3 were upregulated in both knockdown and forced expression of LINC00941, which suggested that LINC00941 may not directly regulate them (Supplementary Figure S2).

English needs to be checked thoroughly.

Response: We made necessary changes.

Reviewer 2 Report

Gist/summary:  The authors delve upon an interesting analysis taking lncRNAs with mitogen activated protein kinase (MAPK) in pancreatic cancer cell lines. In this work, they hypothesize that lncRNAs would be regulated by signaling pathways and may contribute to malignant phenotypes of cancer and thereby finding aberrant signaling pathway in lieu of expression of lncRNAs. The upstream target idnetification, sequencing and cell line studies were meticulously done with inherent statistics. But some points are not clear.

What was the inherent cutoff log2FC  is not clear! Please specify

In performing RNA-Seq twice, one for native fold work after RNAI and the other for idnetifying targeted genes of llncRNAs looks a heavysome work.  Couldn't authors think of bioinformatics analyses?  what was the rationale behind this? Inhernet targets could however be found. 

A methodological flowchart will be  a nice addition. 

Minor but essential

Several propositions like "The"  missing. Ia dded. Pl find attached

Some text appears garbled. pl check 

No space between micro AND RNAs 

The materials an dmethdos setcion on sequencing must be rewritten for brevity: The reads  were aligned  to the UCSC hg38  human RefSeq transcriptome using  HISAT2

NOT USCS

Binidng was mis-spelt

Scores on a scale of 0-5 with 5 being the best 

Language: 3

Novelty: 4

Scope and relevance: 4.5

Brevity: 3

Author Response

Reviewer #2:

Comments and Suggestions for Authors

Gist/summary:  The authors delve upon an interesting analysis taking lncRNAs with mitogen activated protein kinase (MAPK) in pancreatic cancer cell lines. In this work, they hypothesize that lncRNAs would be regulated by signaling pathways and may contribute to malignant phenotypes of cancer and thereby finding aberrant signaling pathway in lieu of expression of lncRNAs. The upstream target identification, sequencing and cell line studies were meticulously done with inherent statistics. But some points are not clear.

What was the inherent cutoff log2FC is not clear! Please specify

Response: We added the following description at line 125: Genes showing two-fold or more changes in expression by MAPK attenuation or LINC00941 knockdown were interpreted as significant.

In performing RNA-Seq twice, one for native fold work after RNAI and the other for identifying targeted genes of lncRNAs looks a heavysome work.  Couldn't authors think of bioinformatics analyses?  what was the rationale behind this? Inhernet targets could however be found. 

Response: We performed RNA seq in 2 stages. 1st was for cells of 3 pancreatic cancer cell lines with or without attenuation of MAPK to identify lncRNAs associated with MAPK. 2nd was for AsPC-1 with or without knockdown of LINC00941 to identify genes modulated by LINC00941 Complete transcriptome data were registered on GEO as stated in methods section as follows: The data discussed in this publication have been deposited in NCBI's Gene Expression Omnibus and are accessible through GEO Series accession number GSE216708 (https://www.ncbi.nlm.nih.gov/geo/query/acc.cgi?acc=GSE216708) and GSE216710 (https://www.ncbi.nlm.nih.gov/geo/query/acc.cgi?acc=GSE216710). We made several bioinformatics analyses described in “RNA-seq analysis” in the method section. We also did additional in-silico analysis of our results to know biological processes of identified genes as listed in Supplementary Table 6.

A methodological flowchart will be a nice addition.

Response: We added the methodological flowchart as Supplementary Figure S1.

Minor but essential

Several propositions like "The" missing. Ia dded. Pl find attached

Response: We corrected our text.

Some text appears garbled. pl check 

Response: We corrected our text.

No space between micro AND RNAs

Response: We corrected our text.

The materials and methods section on sequencing must be rewritten for brevity: The reads were aligned to the UCSC hg38 human RefSeq transcriptome using HISAT2

NOT USCS

Response: We corrected as follows at the sentence at line 147: Reads were aligned to GRCh38 using HISAT2.

Binidng was mis-spelt

Response: We corrected our text.